# Indigenous microorganisms offset the benefits of growth and nutrition regulated by inoculated arbuscular mycorrhizal fungi for four pioneer herbs in karst soil

Yan Sun[1], Muhammud Umer[2], Pan Wu[3], Yun Guo[1], Wenda Ren[1], Xu Han[1], Qing Li[1], Bangli Wu[1], Kaiping Shen[1], Tingting Xia[1], Lipeng Zang[1], Shixiong Wang[1], Yuejun He[1,2]*

**1** Forestry College, Research Center of Forest Ecology, Guizhou University, Guiyang, P. R. China, **2** Institute for Forest Resources & Environment of Guizhou, Guizhou University, Guiyang, P. R. China, **3** Key Laboratory of Karst Georesources and Environment, Ministry of Education, Guizhou University, Guiyang, China

* hyj1358@163.com

**Data Availability Statement:** All relevant data are within the manuscript.

**Funding:** This research was funded by the National Natural Science Foundation of China (NSFC:

## Abstract

Arbuscular mycorrhizal (AM) fungi, as beneficial soil microorganisms, inevitably interact with indigenous microorganisms, regulating plant growth and nutrient utilization in natural habitats. However, how indigenous microorganisms affect the benefits of growth and nutrition regulated by inoculated AM fungi for plants in karst ecosystem habitats remains unclear today. In this experiment, the Gramineae species *Setaria viridis* vs. *Arthraxon hispidus* and the Compositae species *Bidens pilosa* vs. *Bidens tripartita* exist in the initial succession stage of the karst ecosystem. These plant species were planted into different soil microbial conditions, including AM fungi soil (*AMF*), AM fungi interacting with indigenous microorganisms soil (*AMI*), and a control soil without AM fungi and indigenous microorganisms (*CK*). The plant biomass, nitrogen (N), and phosphorus (P) were measured; the effect size of different treatments on these variables of plant biomass and N and P were simultaneously calculated to assess plant responses. The results showed that *AMF* treatment differently enhanced plant biomass accumulation, N, and P absorption in all species but reduced the N/P ratio. The *AMI* treatment also significantly increased plant biomass, N and P, except for the *S. viridis* seedlings. However, regarding the effect size, the AM fungi effect on plant growth and nutrition was greater than the interactive effect of AM fungi with indigenous microorganisms. It indicates that the indigenous microorganisms offset the AM benefits for the host plant. In conclusion, we suggest that the indigenous microorganisms offset the benefits of inoculated AM fungi in biomass and nutrient accumulation for pioneer plants in the karst habitat.

## Introduction

Karst ecosystem occupies approximately 7~12% of emerged land globally, mainly distributed in southwest China, and is characterized by high habitat heterogeneity and high vegetation

31660156; 31360106), the Science and Technology Project of Guizhou Province ([2021] General-455; [2016] Supporting-2805), the Guizhou Hundred-level Innovative Talents Project (Qian-ke-he platform talents [2020] 6004), the First-class Disciplines Program on Ecology of Guizhou Province (GNYL[2017]007), the Talent-platform Program of Guizhou Province ([2017] 5788; [2018]5781), The Basic Research Program in Guizhou Province ([2019]1060). The funders had no role in study design, data collection and analysis, decision to publish, or preparation of the manuscript.

**Competing interests:** NO. The authors have declared that no competing interests exist.

fragmentation [1] with high soil erosion, rocky desertification, and barren vegetation nutrient deficiency [2–7]. However, the karst vegetation retains a robust natural resilience even in harsh habitats [8, 9]. Initially, the pioneer herbaceous plants, mainly from Gramineae and Compositae, have high resistance to drought and barrenness, grow fast, and improve soil structure [10] in the primary succession stage ecosystem restoration [11, 12]. In addition, soil microorganisms play an essential role in recovering degraded karst systems [13] through promoting the growth and nutrient uptake by plants [14, 15] as well as increasing soil nutrient bioavailability [16]. Thus, soil microorganisms in karst vegetation restoration cannot be ignored.

AM fungi, a soil functional microorganism, can play critical roles in recovering degraded terrestrial ecosystems [17]. AM fungi formed a symbiotic relationship with 80% of terrestrial plants [18, 19], improve plant growth, nutrient accumulation [20, 21], enhance drought stress tolerance [22] and maintain soil structure [23], e.g. Guo et al. (2021) [24] proposed that AM fungi differently affected the competitive ability of *Broussonetia papyrifera* and *Carpinus pubescens*; Xia et al. (2020) [25] also showed that AM fungi increased nutrients of host plants by regulating the morphological development of karst plant roots. In addition, Shi et al. (2015) [26] illustrated that AM fungi increased the biomass, N, and P content in shoots and roots of plants. Furthermore, AM fungi mycelium can transfer the photosynthetic carbohydrates from the host plants to the soil, which recruits soil microorganisms [27]. However, we know relatively little about how regulation of plant growth and nutrient by AM fungi is affected by interaction with indigenous microorganisms.

AM fungi via extensive extraradical hyphae interacting with indigenous microbial communities play crucial roles in plant growth in natural habitats [28, 29]. AM fungi and bacteria are ubiquitous in natural soil [30]. Specifically, AM fungi regulate plant growth, and they are positively affected by cooperating with indigenous microorganisms [31, 32] or negatively affected by competing with indigenous microorganisms [33, 34]. Ortiz et al. (2015) [31] suggested that the combination of AM fungi and specific bacteria could promote plant growth by minimizing drought-related stress effects. Artursson et al. (2006) [35] also proposed that the co-inoculation of AM fungi and phosphorus-solubilizing bacteria positively promotes plant nutrient absorption. In addition, plant growth-promoting rhizobacteria could promote mycorrhizal fungal activity and establishment [36–38], which are called mycorrhiza helper bacteria [39]. In contrast, the competition phenomenon between AM fungi and bacteria was also widely reported [40]. Azcón-Aguilar et al. (1997) [41] presented evidence of direct competition between AM fungi and indigenous microorganisms for photosynthetic products of the host plant. Indirectly, Doumbou et al. (2005) [42] proposed that many *Streptomyces* sp. could exude antifungal compounds, which indicated that they are fungal competitors under the appropriate environmental conditions. Thus, the cooperation and competition between AM fungi and indigenous microorganisms are ineluctability in karst soil.

In summary, AM fungi play important roles in improving plant growth and nutrient absorption. However, AM fungi inevitably interact with indigenous microorganisms in the vegetation restoration of the karst-degraded ecosystem. It remains unclear how indigenous microorganisms affect the benefits of growth and nutrition regulated by AM fungi for plants in karst soils. Because of the complexity and uncertainty of the interaction between AM fungi with indigenous microorganisms, it is necessary to assess the effect size of AM fungi and indigenous microorganisms, and their interaction, on plant growth and nutrition. The aim is to clarify how indigenous microorganisms affect the benefits of growth and nutrition regulated by AM fungi for plants in karst soils. We hypothesize that: (1) AM fungi can promote the growth and nutrients of karst plants (H1), according to that AM fungi increased plant biomass and nutrition accumulation [20, 26]. (2) Indigenous microorganisms can offset the benefits

from AM fungi on plant growth and nutrient accumulation (H2), according to that indigenous microorganisms may negatively affect the AM benefits for the host plant through competition [33, 34, 41–43].

## Materials and methods

### Experiment treatments

A potting experiment was conducted by using four herb species: *Setaria viridis*, *Arthraxon hispidus*, *Bidens pilosa*, and *Bidens tripartita* in polypropylene plastic pots in a greenhouse of Guizhou University in Guiyang, China (E: 106˚22′ E; N: 29˚49′ N; 1,120 m above the sea level). Three different microbial conditions soil was created to explore the interaction of AM fungi with indigenous microorganisms in the regulation of plant growth and nutrient utilization. It included AM fungi inoculating into sterilized soil (*AMF* treatment), AM fungi inoculating into natural conditions soil containing indigenous microorganisms (*AMI* treatment), and the control soil by removing microorganisms with sterilization (*CK* treatment). In the beginning, limestone soil (Calcaric regosols, FAO) [44] was collected from a typical karst habitat, from which approximately two-thirds of the soil was used for sterilization at 126˚C, 0.14 Mpa for one hour to eliminate microbes, and one-third of the soil was retained for further experiments. Subsequently, a 2.5 kg soil subsample of the sterilized or unsterilized soil was put into each polypropylene plastic pot (180 mm × 160 mm, diameter × height). Five seeds of *Setaria viridis*, *Arthraxon hispidus*, *Bidens pilosa*, and *Bidens tripartita* were disinfected with a 10% $H_2O_2$ solution for 10 minutes and repeatedly washed with sterile water, and sown in each pot. After sowing seeds in each pot, seeds were covered with 200 g of the respective soil for promoting seed germination. In addition, the sterilized soil was inoculated with 10 g *Glomus mosseae* inoculum as the *AMF* treatment, and the original soil from field habitat was inoculated with 10 g *Glomus mosseae* inoculum as the *AMI* treatment, indicating the AM fungi interacting with the indigenous microorganisms in this experiment. Especially, *CK* treatment received an additional 10ml of the filtrate by weighing 10g of *Glomus mosseae* inoculum with sterile water using a double-layer filter paper, along with a 10 g of sterilized inoculum of *Glomus mosseae* was added in order to maintain the consistency of microflora except for the targeted fungus *Glomus mosseae* corresponding to *AMF* treatment. The inoculum propagated for four months with *Trifolium repens*, including approximately 100 spores per gram soil, hyphae, and colonized root pieces. There is mutual control between two of three treatments: the AM fungi effect through comparing *AMF* with *CK* treatment; the interactive effect of AM fungi with indigenous microorganisms through comparing *AMI* and *CK* treatment; and the indigenous microorganisms effect related to AM fungi through comparing *AMI* with *AMF* treatment. Of course, we had to admit that the unsterilized soil probably had native AM fungi under *AMI* treatment, even the targeted species *Glomus mosseae*. However, it was sure that the *Glomus mosseae* inoculum interacted with native AMF species and indigenous microorganisms; further, they jointly affected plants and soil for growth and nutrition when comparing *AMI* with *AMF*. All treatments were replicated five times, and four plant species contained 60 pots.

The physicochemical properties of limestone soil (per kg) were measured by the methods from Tan (2005) [45], the PH 8.2, total nitrogen (TN) 0.622 g, alkaline hydrolysis nitrogen (AN) 0.315 g, total phosphorus (TP) 1.274 g, available phosphorus (AP) 0.163 g, total potassium (TK) 37.79 g, and available potassium (AK) 0.532 g. All plant seeds were also collected from the same karst habitat used to collect soil. According to the primary field survey, these plants are successive pioneer species of karst communities as the herbaceous stage, which

generally coexist in the same habitat as the main Gramineae and Compositae. Three weeks after seeds germination, only two seedlings were retained in the pot and cultured for five months. All growing seedlings were watered one time per day for maintaining field capacity, then harvested to determine the biomass, N, and P concentrations. The *Glomus mosseae* inoculum was initially purchased from the Institute of Nutrition Resources, Beijing Academy of Agricultural and Forestry Sciences (NO.BGA0046).

## Determinations of the root mycorrhizal colonization, biomass, and the accumulation of nitrogen and phosphorus

The grid line-intersect method determined the root mycorrhizal colonization rate [46]. The biomass of *S. viridis*, *A. hispidus*, *B. pilosa*, and *B. tripartita* were respectively determined by weighing tissue of root, stem, and leaves after drying at 80˚C to constant weight. The nitrogen and phosphorus concentrations in plant tissue were determined by the traditional Kjeldahl method and the Molybdenum-antimony anti-colorimetric method, respectively [47]. Additionally, the accumulations of nitrogen and phosphorus were calculated through nutrient concentration multiplying by biomass, respectively. Then the nutrient accumulation of plant individuals was accumulated by root, stem, and leaf.

## Calculation of effect size

The effect size was calculated using the response ratio (ln*R*) of treatment groups to the control groups plant biomass referred from the proposition of [48] regarding the plant response mycorrhizal fungi. The AM fungi effect (*AME*) by *AMF vs. CK*, the interactive effect of AM fungi with indigenous microorganisms (*AIE*) by *AMI vs. CK*, and the indigenous microorganisms effect related to AM fungi (*IME*) by *AMI vs. AMF* were calculated respectively, due to the mutual control between two of three treatments in this experiment. Therefore, the modified method was adopted according to Hoeksema et al. (2010) [48] and Hedges et al. (1999) [49] as follows:

$$\ln R = \ln (Xt/Xc)$$

Where *Xt* and *Xc* represent the biomass or nutrient accumulation of the plant in the values of the treatment group and control group, respectively, values $> 0$ indicate positive effects promoting plant growth or nutrient accumulation, values $< 0$ indicate negative effects suppressing plant growth or nutrient accumulation.

## Statistical analyses

All of the statistical analyses were performed through SPSS 25.0 software. All of the data were tested for normality and homogeneity of variance before analysis. Two-way ANOVA was applied for assessing the effects of plant species (Ps; *Setaria viridis* vs. *Arthraxon hispidus* vs. *Bidens pilosa* vs. *Bidens tripartita*), soil microbial treatments (Ms; *AMI* vs. *AMF* vs. *CK*), and their interactions (Ms×Ps) on plant biomass, nitrogen accumulation, and phosphorus accumulation, N/P ratio and effect size by the ln*R*. The least significant difference (LSD) test was applied to compare significant differences in root mycorrhizal colonization, biomass, nitrogen, and phosphorus accumulations, and N/P ratio with effect size by the ln*R* among the three different conditions of soil microbial treatments with *AMI*, *AMF*, and *CK* or four plant species of *Setaria viridis* and *Arthraxon hispidus* and *Bidens pilosa* and *Bidens tripartita* at P≤0.05. All graphs were drawn on Origin 2018.

**Table 1. The mycorrhizal colonization rates of *Setaria viridis*, *Arthraxon hispidus*, *Bidens pilosa*, and *Bidens tripartite*.**

| Treatment | Mycorrhizal colonization | | | |
|---|---|---|---|---|
| | *S.viridis* | *A.hispidus* | *B.pilosa* | *B.tripartita* |
| *AMI* | 20.40 ± 0.68cx | 48.60 ± 1.17bx | 65.00 ± 1.48ax | 67.40 ± 1.29ax |
| *AMF* | 18.80 ± 1.43cx | 46.40 ± 1.78bx | 62.20 ± 1.46ax | 64.60 ± 1.17ax |

The different lowercase letters (a, b, c, d) indicate significant differences between plant species of *Setaria viridis*, *Arthraxon hispidus*, *Bidens pilosa*, and *Bidens tripartita* at the 0.05 level; The different lowercase letters (x, y) indicate significant differences between *AMF*, *AMI*, treatments under the same plant.

## Results

### Root mycorrhizal colonization of four plant species under different microbial treatments

A non-significant *AMI* > *AMF* of root mycorrhizal colonization was observed in the four species. However, the root mycorrhizal colonization of *CK* treatment was zero; meanwhile, the AM fungus spore and mycelium were not discovered under *CK* soil substrate via microscopic detection (Table 1). The root mycorrhizal colonization of *B. pilosa* and *B. tripartita* were significantly greater than that of *A. hispidus* and *S. viridis*, respectively, while for *A. hispidus*, it was also greater than *S. viridis*. Besides, there was no significant difference in root mycorrhizal colonization of *B. pilosa* and *B. tripartita* under *AMI* and *AMF* treatments (Table 1). These results indicate root mycorrhizal colonization is species differences, and it provides evidence for host preferences of AM fungal.

### Biomass and its response ratio of four plant species under different microbial treatments

The soil microbial condition treatments (Ms) significantly affected biomass (Table 2). Significantly *AMF* > *AMI* > *CK* of biomass were observed in *A. hispidus*, *B. pilosa*, and *B. tripartita* seedlings except for *S. viridis*. Plant biomass was increased by AM fungus when comparing *AMF* with *CK* and *AMI* with *CK*, respectively. However, the plant biomass under *AMI* was significantly lower than under *AMF* (Fig 1A). The plant species (Ps) also significantly affected individual biomass (Table 2). Under *AMF* and *CK* treatments, the biomass of *A. hispidus* was significantly greater than the other three species. The biomass of *S. viridis* was significantly lower than the other three species under *AMF* and *AMI* treatments. In addition, there was a non-significant difference in biomass observed between *B. pilosa* and *B. tripartita* seedlings under any soil microbial condition treatments (Fig 1A). Meanwhile, the interaction of Ms×Ps

**Table 2. Two-way ANOVAs for the effects of plant species (S. *viridis vs. A. hispidus vs. B. pilosa vs. B. tripartita*) and soil microbial condition (*AMF vs. AMI vs. CK*) on the biomass, the N accumulation, and their response ratio (ln*R*).**

| Factors | df | Biomass | | Response ratio of biomass ($\ln R_{Biomass}$) | | N accumulation | | Response ratio of N ($\ln R_N$) | |
|---|---|---|---|---|---|---|---|---|---|
| | | *F* | *P* | *F* | *P* | *F* | *P* | *F* | *P* |
| Ms | 2 | 124.072 | 0.000*** | 542.979 | 0.000*** | 117.557 | 0.000*** | 416.907 | 0.000*** |
| Ps | 3 | 34.012 | 0.000*** | 59.966 | 0.000*** | 21.133 | 0.000*** | 54.208 | 0.000*** |
| Ms×Ps | 6 | 27.106 | 0.000*** | 33.041 | 0.000*** | 13.492 | 0.000*** | 24.766 | 0.000*** |

Abbreviations: Ms = Soil microbial condition treatments; Ps = Plant species;

* or ** or *** indicates a significant difference in *P* < 0.05 or *P* < 0.01 or *P* < 0.001.

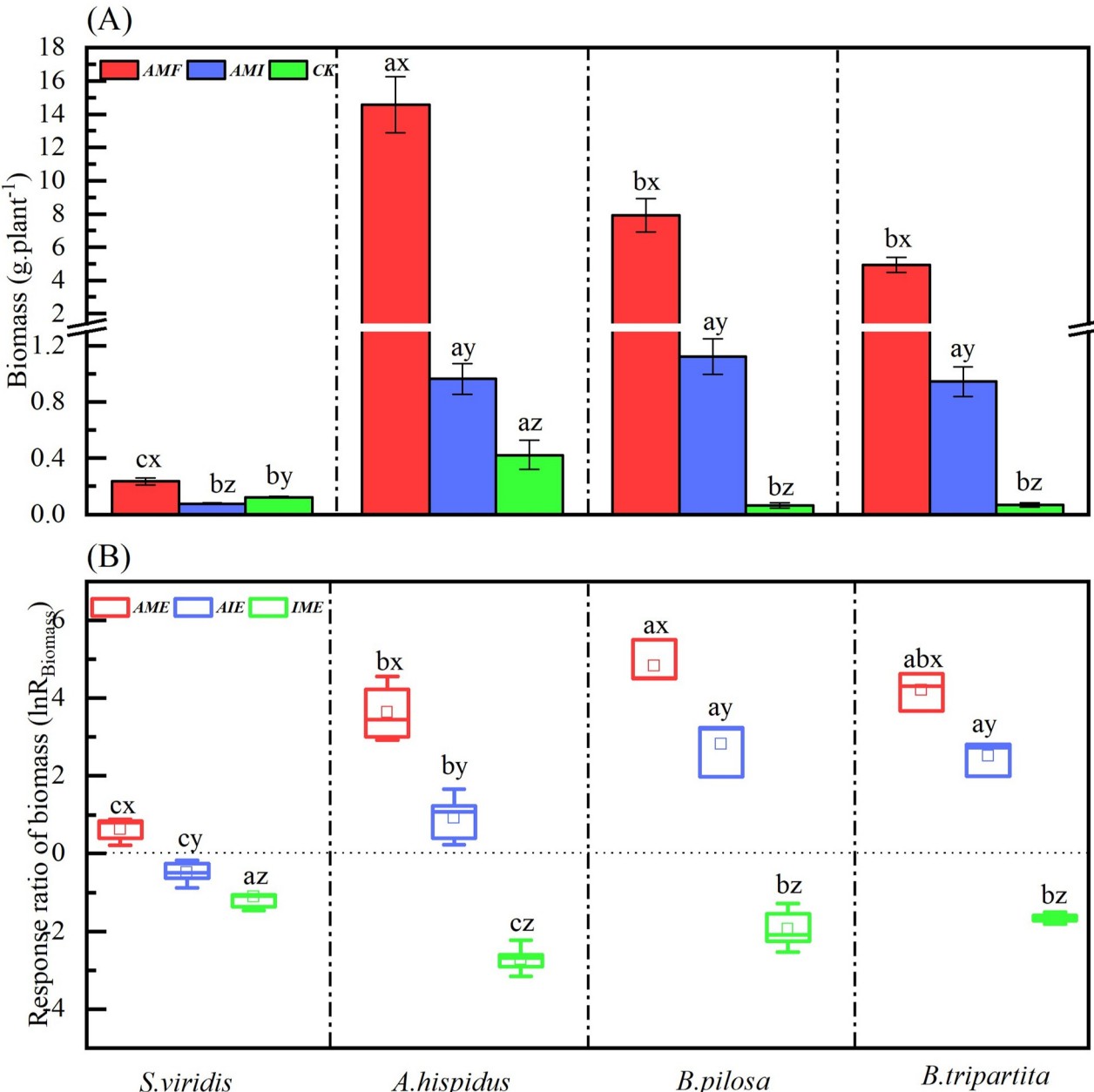

**Fig 1.** The biomass (A) and its response ratio $lnR_{Biomass}$ (B) of four plant species through the different microbial treatments. Abbreviations: *S. v* = *Setaria viridis*; *A. h* = *Arthraxon hispidus*; *B. p* = *Bidens pilosa*; *B. t* = *Bidens tripartita*; *AMF* = the mycorrhizal fungi soil by AM fungi inoculation; *AMI* = the combining soil by AM fungi with indigenous microorganism; *CK* = the sterilized soil as the control by removing microorganism; *AME* = AM fungi effect; *AIE* = interactive effect related to AM fungi interacting with indigenous microbes; *IME* = indigenous microbial effect related to AM fungi. The different lowercase letters (a, b, c, d) indicate significant differences between species under *AMF*, *AMI*, and *CK* treatments, respectively. The different lowercase letters (x, y, z) indicate significant differences between *AMF*, *AMI*, and *CK* treatments for the same species ($P < 0.05$).

significantly affected the individual biomass for the four species (Table 2). The results revealed that *AMF* and *AMI* treatments significantly increased the biomass accumulation of four karst pioneer species. Meanwhile, the biomass was significantly different between *A. hispidus* and *S. viridis* of Gramineae, except for *B. pilosa* and *B. tripartita* under *AMF*.

Similarly, the soil microbial condition treatments (Ms), the plant species (Ps), and their interaction significantly affected the response ratio of biomass ($\ln R_{Biomass}$) (Table 2). On the one hand, a positive effect ($\ln R_{Biomass} > 0$) of biomass was observed in the four species under *AME* and *AIE* conditions except for *S. viridis* in *AIE* (Fig 1B). However, a significant *AME* > *AIE* was observed in $\ln R_{Biomass}$, indicating that AM fungus was beneficial for plant biomass, but the positive effect was decreased when AM fungi interacted with indigenous microorganisms. On the other hand, a negative effect ($\ln R_{Biomass} < 0$) was shown in the *IME* condition (Fig 1B), indicating that indigenous microorganisms offset the AM fungi promotion in plant growth. Precisely, the results indicated that AM fungi significantly increased the biomass accumulation of four karst pioneer species; however, the $\ln R_{Biomass}$ reduction by comparing *AIE* to *AME* specified that the indigenous microorganisms offset the benefits of inoculated AM fungi in promoting plant biomass.

## Nitrogen accumulation and its response ratio of four plant species under different microbial treatments

The soil microbial condition treatments (Ms) significantly affected N accumulation (Table 2). Significantly *AMF* > *AMI* > *CK* of N accumulation were shown in *A. hispidus*, *B. pilosa*, and *B. tripartita* seedlings, except for *S. viridis*. Specifically, the N accumulation was enhanced by AM fungus when comparing *AMF* with *CK* and *AMI* with *CK*, respectively. At the same time, N accumulation under *AMI* was significantly lower than under *AMF* (Fig 2A). The plant species (Ps) also significantly affected N accumulation (Table 2). Under *AMF* and *AMI* treatments, N accumulation in *S. viridis* was significantly lower than other three species. For *CK* treatment, N accumulation in *A. hispidus* was significantly greater than the other three species. Moreover, there was a non-significant difference in N accumulation between *B. pilosa* and *B. tripartita* seedlings under any soil microbial condition treatments (Fig 2A). Furthermore, the interaction of Ms×Ps significantly affected the N accumulation for the four species (Table 2). These results showed that *AMF* and *AMI* treatments significantly increased the N accumulation of four karst pioneer species. Meanwhile, N accumulation was significantly different between *A. hispidus* and *S. viridis*, but not for *B. pilosa* and *B. tripartita* under *AMF*.

Similarly, the soil microbial condition treatments (Ms), the plant species (Ps), and their interaction significantly affected the response ratio of N ($\ln R_N$) (Table 2). One side has a positive effect ($\ln R_N > 0$) of N was observed in four species under *AME* and *AIE* conditions except for *S. viridis* in *AIE* (Fig 2B). However, a significant *AME* > *AIE* was observed in $\ln R_N$, indicating that AM fungus was beneficial for plant N accumulation, but the positive effect was decreased when AM fungi interacted with indigenous microorganisms. Another side has a negative effect ($\ln R_N < 0$) obtainable in the *IME* condition (Fig 2B), indicating that indigenous microorganisms offset the AM fungi promotion in N accumulation. Overall, the results indicated that AM fungi significantly increased the N accumulation of four karst pioneer species; however, the $\ln R_N$ reduction by comparing *AIE* to *AME* specified that the indigenous microorganisms offset the benefits of inoculated AM fungi in promoting N accumulation.

## Phosphorous accumulation and its response ratio of four plant species under different microbial treatments

The soil microbial condition treatments (Ms) significantly affected P accumulation (Table 3). Significantly *AMF* > *AMI* > *CK* of P accumulation was admissible in four species. Unambiguously, AM fungus enhanced P accumulation when comparing *AMF* with *CK* and *AMI* with *CK*; but the P accumulation under *AMI* was significantly lower than under *AMF* (Fig 3A). The plant species (Ps) also significantly affected P accumulation (Table 3). Under *AMF* and *AMI*

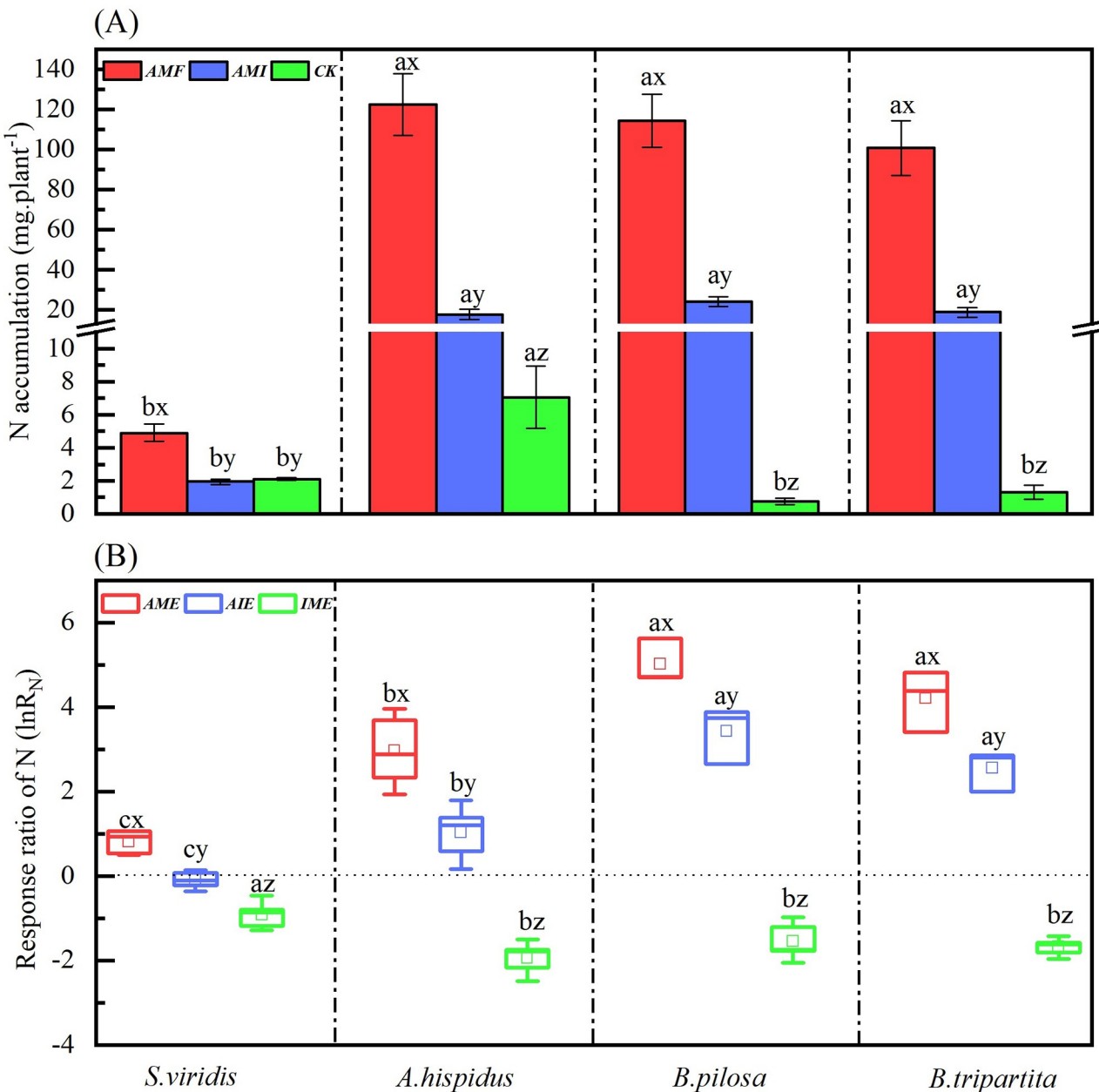

**Fig 2.** N accumulation (A) and response ratio $lnR_N$ (B) of four plant species through the different microbial treatments. The meanings of abbreviations (*S. v*, *A. h*, *B. p* and *B. t*; *AMF*, *AMI*, and *CK*; *AME*, *AIE* and I*ME*) and the lowercase letters (a, b, c, d; x, y, z) are the same as in Fig 1.

treatments, the P accumulation in *S. viridis* was significantly lower than other three species. For *CK* treatments, the P accumulation of *A. hispidus* was significantly greater than the other three species. In addition, there was no significant difference in P accumulation between *B. pilosa* and *B. tripartita* seedlings under any microbial condition soil treatments (Fig 3A). Meanwhile, the interaction of Ms×Ps significantly affected the P accumulation for four species (Table 3). It shows that *AMF* and *AMI* treatments significantly increased the P accumulation of four karst pioneer species. Meanwhile, P accumulation was significantly different between *A. hispidus* and *S. viridis* of Gramineae, except for *B. pilosa* and *B. tripartita* of Compositae under *AMF*.

**Table 3. Two-way ANOVAs for the effects of plant species (*S. viridis vs. A. hispidus vs. B. pilosa vs. B. tripartita*) and soil microbial condition (*AMF vs. AMI vs. CK*) on the P accumulation, the N/P ratio, and their response ratio (ln*R*).**

| Factors | *df* | P accumulation | | Response ratio of P (ln$R_P$) | | N/P ratio | | Response ratio of N/P (ln$R_{N/P}$) | |
|---|---|---|---|---|---|---|---|---|---|
| | | *F* | *P* | *F* | *P* | *F* | *P* | *F* | *P* |
| Ms | 2 | 102.158 | 0.000*** | 394.863 | 0.000*** | 8.263 | 0.000*** | 10.936 | 0.000*** |
| Ps | 3 | 15.069 | 0.000*** | 24.168 | 0.000*** | 20.876 | 0.000*** | 40.158 | 0.000*** |
| Ms×Ps | 6 | 12.138 | 0.000*** | 44.834 | 0.000*** | 9.569 | 0.000*** | 12.175 | 0.000*** |

Abbreviations: Ms = Soil microbial condition treatments; Ps = Plant species;

* or ** or *** indicates a significant difference in $P < 0.05$ or $P < 0.01$ or $P < 0.001$.

Likewise, the soil microbial condition treatments (Ms), the plant species (Ps), and their interaction significantly affected the response ratio of P (ln$R_P$) (Table 3). Alternatively, it has a positive effect (ln$R_P > 0$) of P on four species under *AME* and *AIE* conditions. However, a significant *AME > AIE* was observed in ln$R_P$, indicating that AM fungus was beneficial for plant P accumulation, but the positive effect was decreased when AM fungi interacted with indigenous microorganisms (Fig 3B). It also has a negative effect (ln$R_P < 0$) in the *IME* condition and depicts that the indigenous microorganisms offset the AM fungi promotion in P accumulation. Therefore, the results consolidated that AM fungi significantly increased P accumulation of four karst pioneer species, then the ln$R_P$ reduction by comparing *AIE* to *AME* designated that the indigenous microorganisms offset the benefits of inoculated AM fungi in promoting P accumulation.

### N/P ratio and its response ratio of four plant species under different microbial treatments

The soil microbial condition treatments (Ms) significantly affected the N/P ratio (Table 3), significantly greater N/P ratio between plant species ranked as the *CK > AMF ≈ AMI* for *S. viridis*, the *AMI > CK ≈ AMF* for *A. hispidus*, the *AMI > AMF > CK* for *B. Pilosa*, and *CK ≈AMI > AMF* for *B. tripartita* (Fig 4A). The plant species (Ps) also significantly affected the N/P ratio (Table 3), and the N/P ratio for four plants showed species differences under different soil microbial treatments. Explicitly, there was a non-significant difference in the N/P ratio of the four species under *AMF* treatments. Under *AMI* treatments, the N/P ratio of *A. hispidus* and *B. tripartita* were significantly greater than *S. viridis* and *B. pilosa*, respectively. In the interim, the N/P ratio of the *B. pilosa* was greater than *S. viridis* seedlings. Under *CK* treatment, the N/P ratio of *B. tripartita* was significantly greater than the other three species, while the N/P ratio of *B. pilosa* was significantly lower than the other three species (Fig 4A). Likewise, the interaction of Ms×Ps significantly affected the N/P ratio for four species (Table 3). Therefore, AM fungi significantly reduced the N/P ratio of four species. Equally, the soil microbial condition treatments (Ms), the plant species (Ps), and their interaction significantly affected the response ratio of N/P (ln$R_{N/P}$) (Table 3, Fig 4B). Overall, AM fungi significantly reduced the N/P ratio for the four-karst pioneer species, portraying that the AM fungi alleviate P limitation and promote plant growth in karst areas with low P.

## Discussion

### AM fungi differently regulated the plant growth and nutrient accumulation

AM fungi significantly increased biomass and N and P accumulation for the four karst pioneer species (Figs 1A, 2A and 3A). Consistently, the positive influence of AM fungi inoculation on host plant growth and nutrient accumulation was also observed in some previous studies

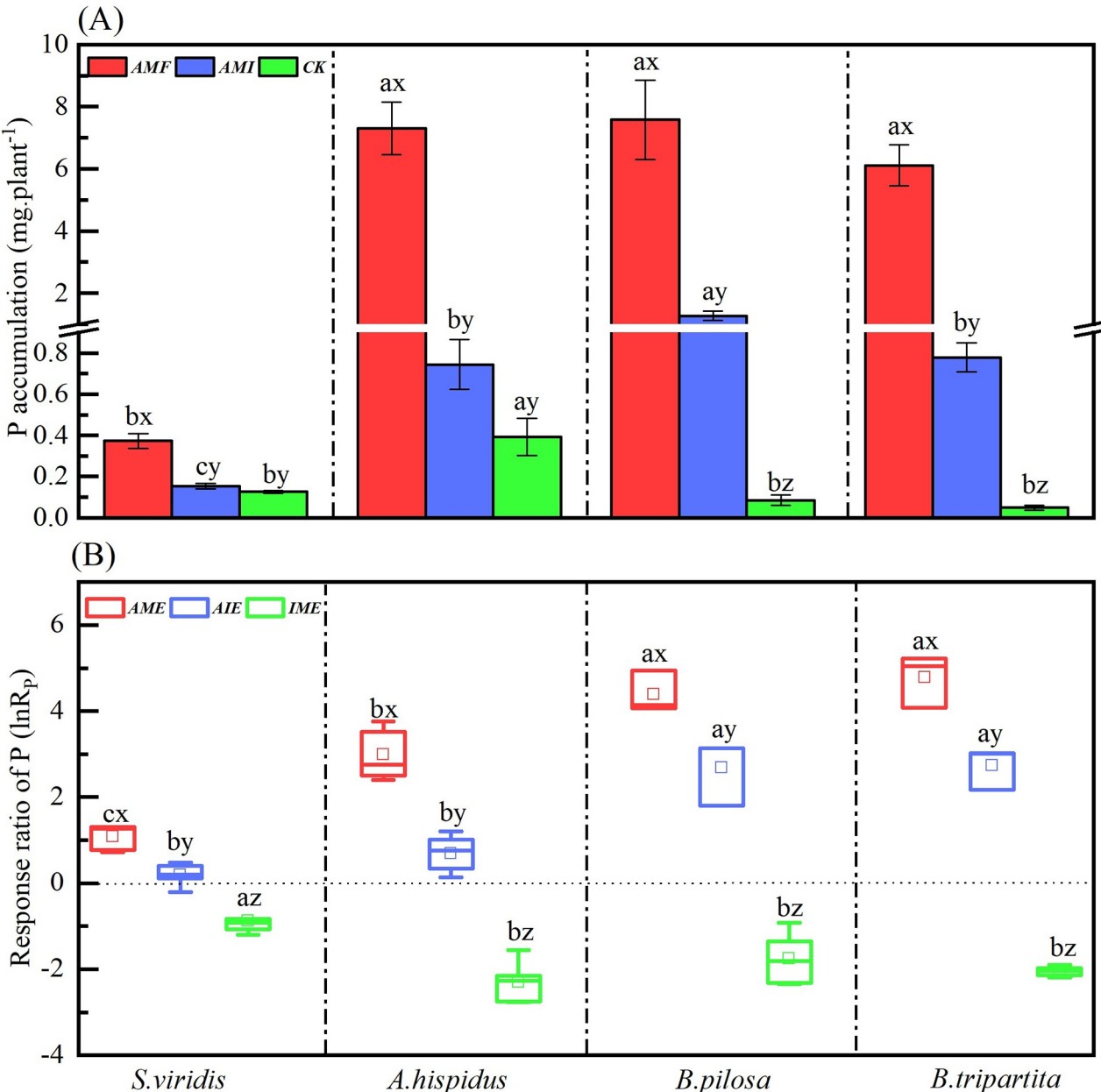

**Fig 3.** P accumulation (A) and response ratio *lnR_P* (B) of four plant species through the different microbial treatments. The meanings of abbreviations (*S. v*, *A. h*, *B. p* and *B. t*; *AMF*, *AMI*, and *CK*; *AME*, *AIE* and I*ME*) and the lowercase letters (a, b, c, d; x, y, z) are the same as in Fig 1.

[50, 51]. For instance, He et al. (2017) [20] showed that AM fungi enhanced plant growth and nutrient absorption of *B. papyrifera* and *B. pilosa* in karst soil, which is consistent with our results that AM fungi significantly increased biomass and accumulation of N and P for the four plants. There are two main mechanisms that AM fungi promote plant growth and nutrient accumulation. One side is that AM fungi can extend the absorbing network beyond the rhizosphere nutrient depletion region and absorb a larger amount of soil mineral nutrients, thereby improving the ability of plants to obtain nutrients [52] and ultimately benefit plant growth [53–55]. Another is that AM fungi can secrete organic acids and soil enzymes to

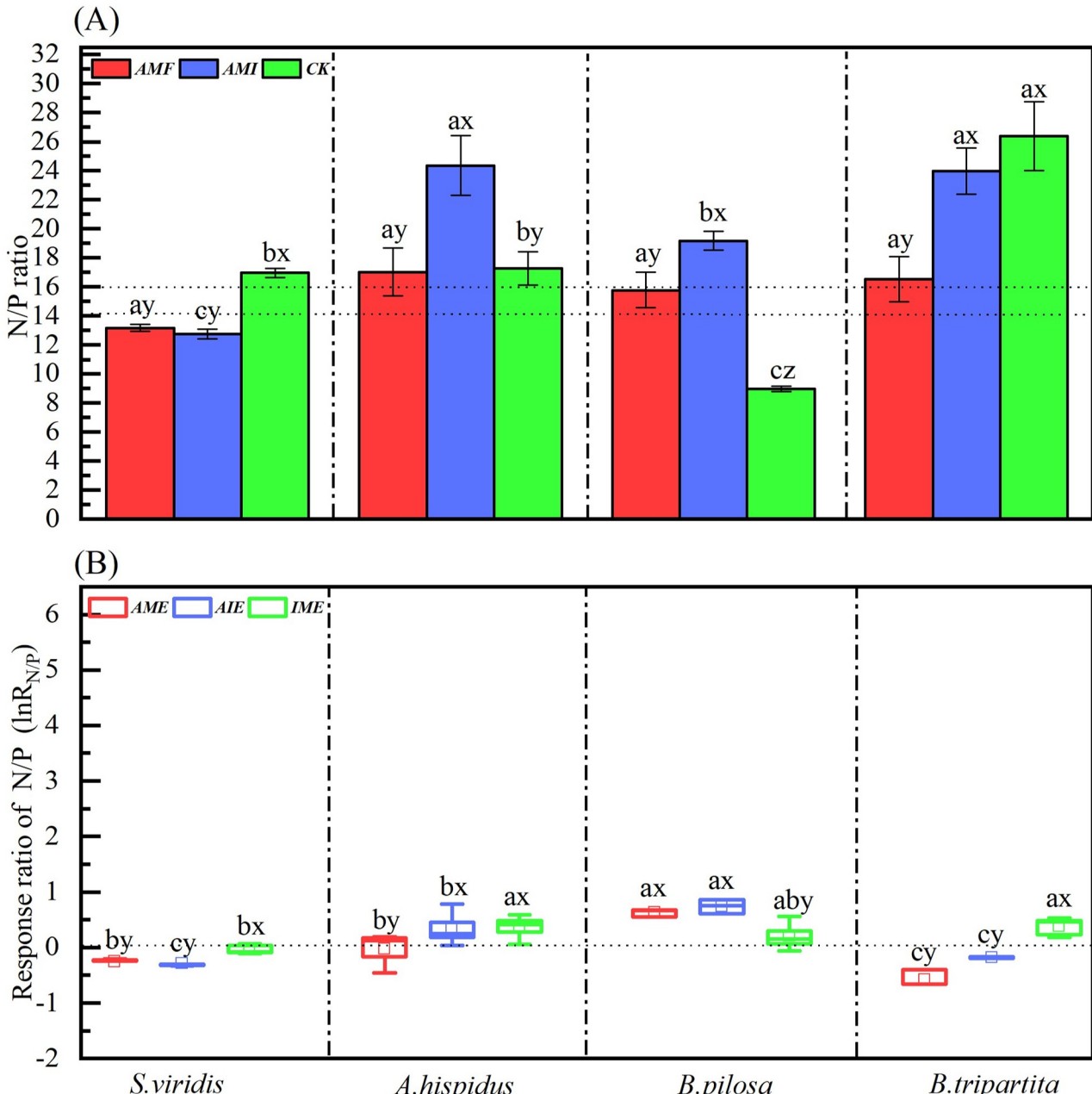

**Fig 4.** N/P ratio (A) and response ratio $lnR_{N/P}$ (B) of four plant species through the different microbial treatments. The meanings of abbreviations (*S. v*, *A. h*, *B. p* and *B. t*; *AMF*, *AMI*, and *CK*; *AME*, *AIE* and I*ME*) and the lowercase letters (a, b, c, d; x, y, z) are the same as in Fig 1.

dissolve the insoluble nutrients and mineralize the organic nutrient [56–58], thereby promoting the availability of soil nutrients [59]. Elbon and Whalen (2014) [60] illustrated that AM fungi could increase the plant-available P concentration by secreting organic acids and phosphatase enzymes. Therefore, AM fungi facilitated the growth and nutrient accumulation of four karst pioneer plants, which can verify the hypothesis of H1. However, the specific mechanism of AM fungi affecting nutrient accumulation of karst pioneer species needs to be explored further.

The N/P ratio can predict plant nutrient restrictions [61]. A low N/P ratio ($< 14$) indicates N limitation, whereas a high N/P ratio ($> 16$) indicates P limitation, and both N and P limit plant growth when the N/P ratio is between 14 and 16 [62]. In our experiment, the N/P ratio of all species was greater than 16 under *AMI* and *CK* treatments, except for *S. viridis* under *AMI* and *B. pilosa* under *CK*, showing that plant growth was mainly limited by phosphorus in karst soil. However, the N/P ratio of the four species significantly decreased under *AMF* treatments compared with *AMI* and *CK* treatments for a whole (Fig 4A). AM fungi reduced the N/P ratio of seedlings, representing that AM fungus is more effective in assisting plants in obtaining P than N by alleviating P limitation. These results were similar to those of Shen et al. (2020) [63], who suggested that AM fungi alleviated the P limitation of plants via the mycorrhizal network in low-P karst soils. Consequently, the AM fungi play a vital role in alleviating the nutritional restriction of nutrient-deficient karst soils.

AM fungi enhanced four plants' biomass, N, and P accumulation differently. Meanwhile, the *A. hispidus*, *B. pilosa*, and *B. tripartita* obtained greater benefits than the *S. viridis* (Figs 1A, 2A and 3A), demonstrating that the promotion effect of AM fungi on plants was different by host type. Besides, the mycorrhizal colonization of *A. hispidus*, *B. pilosa*, and *B. tripartita* was significantly higher than *S. viridis* (Table 1). It was well proof of the different roles of AM fungi on different species, and these differences reflected that AM fungi had the selectivity for host plants. AM fungi showed host-specific growth response [64] and induced differential growth responses in host plant species [65]. It was similar to the research conducted by Liu et al. (2003) [66], who proposed that *Nicotiana tabacum* was a more favorable host plant for *Glomus constrictum* and *Glomus multicaule* to the other hosts. Therefore, AM fungi are crucial for plant growth and nutrient utilization. However, the mutual selection between AM fungi and host plants cannot be ignored, and thus the specific mechanism of selective plant-AMF combinations of karst pioneer species needs to be explored in further study.

## Indigenous microorganisms affected the benefits of AM fungi on plant growth and nutrient accumulation

In this experiment, the positive AM fungi effect on plant growth and nutrition was greater than the interactive effect related to AM fungi interacting with indigenous microorganisms for a whole (Figs 1B, 2B and 3B). It seems to imply that the indigenous microorganisms offset the benefits of AM fungi on plant growth and nutrient accumulation, signifying a negative relationship between AM fungi and indigenous microorganisms. Previous studies have demonstrated that AM fungi interact with a wide variety of indigenous microorganisms [67, 68]. Meanwhile, AM fungi regulated plant growth positively affected by cooperating with indigenous microorganisms [32] or negatively affected by competing with indigenous microorganisms [34], which depended on the species of indigenous microorganisms that interact with AM fungi [69–71]. Positively, Mortimer et al. (2012) [72] presented a synergistic relationship between AM fungi and nitrogen-fixing bacteria showing additive benefits for the growth and nutrient accumulation in the *Acacia cyclops*. Artursson et al. (2006) [35] illustrated that the plant growth-promoting rhizobacteria (PGPR) could enhance the activity of AM during a symbiotic relationship with the host plant. It is because of the stimulatory effects of PGPR on AM growth [73]. Negatively, AM fungi can compete with indigenous microorganisms to produce different effects on plant growth [74]. Some bacteria in the rhizosphere would compete for resources with AM fungi or inhibit the activity of AM fungi, thereby affecting plant growth [75]. It is because indigenous microorganisms have great advantages in colonizing plant roots due to their priority in resources and allocating root space of the host plants compared with colonizers [76, 77]. In addition, Dąbrowska et al. (2014) [78] presented that inoculation AM

fungi promoted the growth of plants, but interactive effects of AM fungi with indigenous microorganisms inhibited plant growth. It was similar to our study that AM fungi positively affected plant growth and nutrient accumulation; however, indigenous microorganisms reduced this effect, indicating a negative relationship between AM fungi and indigenous microorganisms. It is possibly caused by the competition between AM fungi and indigenous microorganisms, mainly two sides. One side is interference competition, meaning that some microbes directly inhibit the function of AM fungi via exuding allelochemical substances [79] and bacterial antibiotics [80, 81]. For example, Doumbou et al. (2005) [42] proposed that numerous *Streptomyces* sp. could exude antifungal compounds, thereby inhibiting the function of AM fungi under certain environmental conditions. The other side is resources, and ecological niches competition, which was proposed by Leigh et al. (2011) [43] who suggested that resource competition for decomposition products between AM fungi and bacteria, resulting in an antagonistic relationship between them. Niwa et al. (2018) [76] suggested that the fungus inoculum mainly competed with the indigenous fungi, probably because their life-history strategy was identical to the inoculum fungus. All the above-mentioned can explain why the indigenous microorganisms relieved the benefits of AM fungi on plant growth and nutrient accumulation. It was consistent with Biró et al. (2000) [82], who found the indigenous microflora greatly reduced the functioning of the functioning of the mycorrhizal inoculum. Collectively, indigenous microorganisms offset the benefits of AM fungi in this study, which illustrated the interactions between AM fungi and indigenous microorganisms in karst areas should be mainly a negative relationship, it verified the hypothesis of H2 that indigenous microorganisms offset the benefits of AM fungi on plant growth and nutrient accumulation. However, the specific mechanisms of the negative relationship between specific microorganisms and AM fungi in karst soil remain to be further studied.

## Conclusions

In this experiment, AM fungi significantly enhanced the biomass, N, and P accumulation for the four species but reduced the N/P ratio partly. AM fungi interacting with indigenous microorganisms increased plant biomass, N, and P accumulation, except for *S. viridis* seedlings. However, the benefits from interaction were lower than benefits from AM, indicating that the indigenous microorganisms offset the benefits of AM fungi for host plants. In conclusion, we suggest that the indigenous microorganisms offset the benefits of growth and nutrition regulated by inoculated AM fungi for pioneer plants in karst soil. Finally, it is necessary to understand the interactions of AM fungi with indigenous microbial communities to better apply mycorrhizal technology to the degraded ecosystem in karst areas.

## Acknowledgments

We thank Xinyang Xu, Lu Gao, Li Wang, Xiaorun Hu and Jingting Li for helping in this experiment. We are grateful to the Institute of Nutrition Resources, Beijing Academy of Agricultural and Forestry Sciences for providing *Glomus mosseae* (NO. BGA0046) for use in our experiments.

## Author Contributions

**Conceptualization:** Xu Han, Yuejun He.

**Data curation:** Qing Li, Bangli Wu, Kaiping Shen, Tingting Xia.

**Formal analysis:** Yan Sun, Lipeng Zang, Shixiong Wang.

**Funding acquisition:** Yuejun He.

**Methodology:** Xu Han, Yuejun He.

**Project administration:** Yuejun He.

**Supervision:** Yuejun He.

**Writing – original draft:** Yan Sun, Yun Guo, Wenda Ren.

**Writing – review & editing:** Yan Sun, Muhammud Umer, Pan Wu, Yuejun He.

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
