## [Decision Letter · Decision Letter 0]

18 Jan 2022

PONE-D-21-40761Indigenous microorganisms relieved the benefits of growth and nutrition regulated by arbuscular mycorrhizal fungi for four pioneer herbs in karst soilPLOS ONE

Dear Dr. He,

Thank you for submitting your manuscript to PLOS ONE. After careful consideration, we feel that it has merit but does not fully meet PLOS ONE’s publication criteria as it currently stands. Therefore, we invite you to submit a revised version of the manuscript that addresses the points raised during the review process.

ACADEMIC EDITOR: The study is interesting and the manuscript is relatively well written. But the manuscript still have some problems as suggested by the reviewers. The authors should respond to the comments of the reviewers one by one and revise the manuscript accordingly.  

We look forward to receiving your revised manuscript.

Kind regards,

Jian Liu

Academic Editor

PLOS ONE

Journal Requirements:

Reviewers' comments:

Reviewer's Responses to Questions

**Comments to the Author**

1. Is the manuscript technically sound, and do the data support the conclusions?

Reviewer #1: Yes

Reviewer #2: Yes

Reviewer #3: Yes

2. Has the statistical analysis been performed appropriately and rigorously? 

Reviewer #1: Yes

Reviewer #2: Yes

Reviewer #3: Yes

3. Have the authors made all data underlying the findings in their manuscript fully available?

Reviewer #1: Yes

Reviewer #2: Yes

Reviewer #3: Yes

4. Is the manuscript presented in an intelligible fashion and written in standard English?

Reviewer #1: No

Reviewer #2: Yes

Reviewer #3: Yes

5. Review Comments to the Author

Reviewer #1: In this manuscript, the authors explored and discussed the interactions between Arbuscular mycorrhizal and indigenous microorganisms in regarding to their effects on plant growth and nutrient accumulation. The findings may help elucidate the role of AMF and other soil microorganisms in constructing the plant communities in the Karst area in China.

Main suggestions;

1. ‘Relieve’ usually refers to lightening the pressure, stress, weight, etc. on (something)(https://www.collinsdictionary.com/us/dictionary/english/relieve), which is the bad situation of something.

Therefore, ‘offset’ is recommend here to replace ‘relieve’.

The definition of ‘offset’ is something that counterbalances, counteracts, or compensates for something else; compensating equivalent

(https://www.collinsdictionary.com/us/dictionary/english/offset).

2. Th title is suggested as ‘Indigenous microorganisms relieved the benefits of growth and nutrition regulated by inoculated arbuscular mycorrhizal fungi for four pioneer herbs in karst soil’，as the the native soil microbes and native AMF were not separated in this study.

3. There are too many English grammar mistakes in the manuscript. It is strongly suggested that the English expresses should be checked through the whole manuscript. Some corrections were made in the manuscript.

4. Line 353: the treatment description is not consistent with the Methods part.

5. In the Discussion part, there are too much discussions on the effects of AMF, which were already intensively studied by other researchers. Furthermore, it is better to extend the findings of this study to the mechanisms of ecological processes in the Karst area or how this findings can be applied in the restoration of the vegetation in the Karst area.

Reviewer #2: This article entitled "Indigenous microorganisms relieved the benefits of growth and nutrient regulated arbuscular mycorrhizal fungi for four pioneer herbs in karst soil", provides an interesting work about the effects of mycorrhizal fungi interacting with indigenous microorganisms on plants in degraded soil. The authors claimed that the indigenous microorganisms relieved the benefits of AM fungi in the growth and nutrient absorption of four plants in kast. The topic is very interesting and innovative. The experiment is well done and the writing is good. Some modifications are necessary before the consideration of publication.

In general:

1. How can you give the H2 “Indigenous microorganisms relieved the benefits of AM fungi on 98 plant growth and nutrient accumulation”? it is not enough based your literatures review to deduce this H2.

2. Why you chose the four species to manipulate the experiment? Please give the reason.

3. In the discussion you paid more attention on the effect of AMF on plant growth and nutrient absorption. However, I think the combined effects of mycorrhizal fungi and indigenous microorganisms is more important to explanation.

Some details:

1. Line 36: indigenous microbes are inconsistent with line 28.

2. Line 106-108: do you sure the consistency of soil condition in physicochemical properties in AMF, AMI and CK? It is different in natural soil and sterilized soil in general cognition except for microbes.

3. Line 118: 10 g Glomus mosseae should being 10 g Glomus mosseae inoculum.

4. Line 118-119: Did the 10 g Glomus mosseae inoculum include the spore, hyphal and root piece? Please give the information.

5. Line 152: are you sure this condition is the constant weight of drying?

6. Line 173-174: Whether the data has been tested for normality and homogeneity of variance before analysis? In the best way, additional description is necessary to ensure the feasibility of statistical data.

7. Line 154-156: how to calculate the accumulations, can you give us the details about it?

8. Line 290-291: Change "in negative N/P of " for "in negative N/P ratio of "; Change "in positive N/P of " for "in positive N/P ratio of ";

9. Line 322: Change " the N/P of " for "the N/P ratio of "

10. Line 272-273: the indigenous microorganisms relieved the benefits of AM fungi on P accumulation. This sentence is unclear and contradicts the first part of the sentence (AM fungi improved P accumulation).

11. Line 376: Change "streptomycetes " for " Streptomyces sp."

Reviewer #3: Soil microbial interactions play an important role for plant adaptation in natural habitat. As a kind of beneficial microorganisms, Arbuscular mycorrhizal fungi largely promote growth via the improvement of mineral nutrients for the host plant. This paper attempts to solve the interaction between AM fungi and indigenous microorganisms and explore the benefits of indigenous microorganisms on AM fungi promoting plant growth and nutrient utilization through four karst herbs, which were planted in three different microbial condition soil. The results indicated that the indigenous microorganisms relieved AM fungi's benefits in biomass and nutrient accumulation for plants. I believe this work is interesting and meaningful to apply mycorrhizal technology for restoring in degraded karst areas. However, it still needs to improve in some points as the potential publication of this paper, in detail as follows:

1.Line 92-95: This sentence of “Thus, an experiment was ……with indigenous microorganisms”, is not necessary in the Introduction section. It is better to take it into the Methods section.

2.Line 103: do the “1120m.a.s.l” represent elevation? Please correct it.

3.Line 104: “ soil microbial conditions” should be “ soil microbial condition soil”.

4.Line 110: specify limestone soil as International Soil Classification

5.Line 117-118: I confused the reason about promoting germination rate by yours treatment of 200g soil. Please check it and clear it.

6.Line 121-122: This does not makes sense at all. Are you saying that you added AMF inoculum to your treatment control? If so, that does not constitute a control at all.

7.Line 121-123 This part (starting from "Especially, a 10 g…" and ending on "… a double-layer filter paper") is not clear at all. Please make it clear.

8.Line 302: please correct the citation of He, Jiang et al.(2017).

9.Line 328: please correct the citation of Shen, Cornelissen et al.(2017). Check all reference citations in full text, I think it's not standard.

10.Line 332-335: This sentence was so long, I'm very confused with this result; please make it clear and shorten it.

11.Lin 399: change “we can say that” being “ we suggest that”, delete “Finally”.

12.The discussion needs further refinement and accuracy, comparing your results with previous researches for drawing relevant conclusions.

6. PLOS authors have the option to publish the peer review history of their article (what does this mean?). If published, this will include your full peer review and any attached files.

Reviewer #1: No

Reviewer #2: No

Reviewer #3: No

---

## [Author Response · Author response to Decision Letter 0]

2 Feb 2022

Journal Requirements

Q1: Please ensure that your manuscript meets PLOS ONE's style requirements, including those for file naming. The PLOS ONE style templates can be found at https://journals.plos.org/ plosone/s/file?id=wjVg/PLOSOne_formatting_sample_main_body.pdf and https://journals.plos.or

g/ plosone/s/file?id=ba62/PLOSOne_formatting_sample_title_authors_affiliations.pdf

RESPONSE: Thank you for your comments. We revised the manuscript to meet PLOS ONE's style requirements.

Q2: We note that the grant information you provided in the ‘Funding Information’ and ‘Financial Disclosure’ sections do not match. When you resubmit, please ensure that you provide the correct grant numbers for the awards you received for your study in the ‘Funding Information’ section.

RESPONSE: Thank you for your comments. I checked and ensured that we provide the correct grant numbers for the awards you received for your study in the ‘Funding Information’ section.

Q3: Please review your reference list to ensure that it is complete and correct. If you have cited papers that have been retracted, please include the rationale for doing so in the manuscript text, or remove these references and replace them with relevant current references. Any changes to the reference list should be mentioned in the rebuttal letter that accompanies your revised manuscript. If you need to cite a retracted article, indicate the article’s retracted status in the References list and also include a citation and full reference for the retraction notice.

RESPONSE: Thank you for your comments. 

In the revised version: (1) In the materials and methods section, we added a reference [43], see Line106.

(2) In the discussion section, we added four references [78], [80], [81] and [82], see Line 364-365 and Line 367-371.

[43] He YJ, Cornelissen JHC, Wang P, Dong M, and Ou J. Nitrogen transfer from one plant to another depends on plant biomass production between conspecific and heterospecific species via a common arbuscular mycorrhizal network. Environmental Science and Pollution Research. 2019;26(9):8828-8837. https://doi.org/10.1007/s11356-019-04385-x.

[78] Bender SF, Schlaeppi K, Held A, and Van der Heijden MGA. Establishment success and crop growth effects of an arbuscular mycorrhizal fungus inoculated into Swiss corn fields. Agriculture Ecosystems & Environment. 2019;273:13-24. https://doi.org/10.1016/j.agee.2018.12.003.

[80] Niwa R, Koyama T, Sato T, Adachi K, Tawaraya K, Sato S, et al. Dissection of niche competition between introduced and indigenous arbuscular mycorrhizal fungi with respect to soybean yield responses. Scientific Reports. 2018;8. https://doi.org/10.1038/s41598-018-25701-4.

[81] Hausmann NT and Hawkes CV. Order of plant host establishment alters the composition of arbuscular mycorrhizal communities. Ecology. 2010;91(8):2333-2343. https://doi.org/10.1890/09-0924.1.

[82] Dąbrowska G, Baum C, Trejgell A, and Hrynkiewicz K. Impact of arbuscular mycorrhizal fungi on the growth and expression of gene encoding stress protein–metallothionein BnMT2 in the non‐host crop Brassica napus L. J. Plant Nutr. Soil Sci. 2014;177(3):459-467. https://doi.org/10.1002/jpln.201300115.

Reviewer #1:

Q1: In this manuscript, the authors explored and discussed the interactions between Arbuscular mycorrhizal and indigenous microorganisms in regarding to their effects on plant growth and nutrient accumulation. The findings may help elucidate the role of AMF and other soil microorganisms in constructing the plant communities in the Karst area in China.

RESPONSE: Thank you for your comments. According to your suggestions, we revised the title of the paper to ‘Indigenous microorganisms offset the benefits of growth and nutrition regulated by inoculated arbuscular mycorrhizal fungi for four pioneer herbs in karst soil’, and revised the conclusion in abstract, result, and conclusion section. 

Q2: ‘Relieve’ usually refers to lightening the pressure, stress, weight, etc. on (something)(https://www.collinsdictionary.com/us/dictionary/english/relieve), which is the bad situation of something. Therefore, ‘offset’ is recommend here to replace ‘relieve’. The definition of ‘offset’ is something that counterbalances, counteracts, or compensates for something else; compensating equivalent (https://www.collinsdictionary.com/us/dictionary/english/offset).

RESPONSE: Thanks a lot for your good suggestions. We have modified ‘relieve’ to ‘offset’ all in the revised manuscript. see Line36, Line40, Line93, Line216, Line219, Line242, Line245, Line268, Line271 and Line401 of the revision.

Q3: The title is suggested as ‘Indigenous microorganisms relieved the benefits of growth and nutrition regulated by inoculated arbuscular mycorrhizal fungi for four pioneer herbs in karst soil’，as the the native soil microbes and native AMF were not separated in this study.

RESPONSE: Thank you for your good suggestions. We have modified the title of the article to ‘Indigenous microorganisms offset the benefits of growth and nutrition regulated by inoculated arbuscular mycorrhizal fungi for four pioneer herbs in karst soil’, and revised the conclusion in abstract, result, and conclusion section.

Q4: There are too many English grammar mistakes in the manuscript. It is strongly suggested that the English expresses should be checked through the whole manuscript. Some corrections were made in the manuscript.

RESPONSE: Many thanks for your good comments. We further checked and modified the language and refined expression to the whole manuscript in the new version.

Q5: Line 353: the treatment description is not consistent with the Methods part.

RESPONSE: Thank you for your comments and questions. We checked and corrected it in Line 350 of the revised version.

Q6: In the Discussion part, there are too much discussions on the effects of AMF, which were already intensively studied by other researchers. Furthermore, it is better to extend the findings of this study to the mechanisms of ecological processes in the Karst area or how this findings can be applied in the restoration of the vegetation in the Karst area.

RESPONSE: Thank you for your good suggestions. We widely agree with your view that there are too many discussions on the effects of AMF, and it is better to extend the findings of this study to apply them in the restoration of the vegetation in the Karst area. We have checked and revised the Discussion section and Conclusion section carefully and deeply. In order to highlight the emphasis of this paper is not only on the effects of AMF, but we have also enriched the content of the interaction of AM fungi and indigenous microorganisms, and then compare with others, as follows:

(1) AM fungi regulated plant growth positively affected by cooperating with indigenous microorganisms or negatively affected by competing with indigenous microorganisms. When we discussed‘Negatively affected’section, we added the sentence ‘AM fungi can compete with indigenous microorganisms to produce different effects on plant growth, and clarified possible reason for this occurrence. ‘It is because indigenous microorganisms have great advantages in colonizing plant roots due to their priority in resources and allocating root space of the host plants compared with colonizers’, and the compare with ours. See the specific explanation of Line 364-365 and Line 367-371 in the new version. 

(2) In addition, in the Conclusion section, based on the results of this study, we extended mycorrhizal technology to the degraded ecosystem in karst areas, see Line 402-404 of the revised version.

Reviewer #2:

Q1: This article entitled "Indigenous microorganisms relieved the benefits of growth and nutrient regulated arbuscular mycorrhizal fungi for four pioneer herbs in karst soil", provides an interesting work about the effects of mycorrhizal fungi interacting with indigenous microorganisms on plants in degraded soil. The authors claimed that the indigenous microorganisms relieved the benefits of AM fungi in the growth and nutrient absorption of four plants in kast. The topic is very interesting and innovative. The experiment is well done and the writing is good. Some modifications are necessary before the consideration of publication.

RESPONSE: Thank you for your comments. We have completely revised the manuscript in the new version. 

In the revised version: (1) According to the suggestions of Reviewer #1, we revised the title of the paper to ‘Indigenous microorganisms offset the benefits of growth and nutrition regulated by inoculated arbuscular mycorrhizal fungi for four pioneer herbs in karst soil’, and revised the conclusion in abstract, result and conclusion section.

(2) In order to better present the results of this paper, we added some examples about the relationship between AM fungi and indigenous microorganisms in the discussion section to combine the explanation, see Line 364-365 and Line 367-371 of the revision.

Q2: How can you give the H2 “Indigenous microorganisms relieved the benefits of AM fungi on 98 plant growth and nutrient accumulation”? it is not enough based your literatures review to deduce this H2.

RESPONSE: Thanks a lot for your good suggestions.

In the revised version: (1) According to the comments of Reviewer #1, we modified H2 to “Indigenous microorganisms offset the benefits of inoculated AM fungi on plant growth and nutrient accumulation”. (2) We have added appropriate discussion in revied version to deduce H2“Indigenous microorganisms offset the benefits of inoculated AM fungi on plant growth and nutrient accumulation”, by that AM fungi can compete with indigenous microorganisms to produce different effects on plant growth [1] ……. it is because indigenous microorganisms have great advantages in colonizing plant roots due to their priority in resources and allocating root space of the host plants compared with colonizers [2,3]. In addition, Dąbrowska et al. (2014) [4] presented that inoculation AM fungi promoted the growth of plants, but in the soil with indigenous microorganisms, growth inhibition after inoculation was observed compared to the control. It was similar to our study that……; see Line 364-365 and Line 367-371 of revised version.

[1] Bender SF, Schlaeppi K, Held A, and Van der Heijden MGA. Establishment success and crop growth effects of an arbuscular mycorrhizal fungus inoculated into Swiss corn fields. Agriculture Ecosystems & Environment. 2019;273:13-24. https://doi.org/10.1016/j.agee.2018.12.003.

[2] Niwa R, Koyama T, Sato T, Adachi K, Tawaraya K, Sato S, et al. Dissection of niche competition between introduced and indigenous arbuscular mycorrhizal fungi with respect to soybean yield responses. Scientific Reports. 2018;8. https://doi.org/10.1038/s41598-018-25701-4.

[3] Hausmann NT and Hawkes CV. Order of plant host establishment alters the composition of arbuscular mycorrhizal communities. Ecology. 2010;91(8):2333-2343. https://doi.org/10.1890/09-0924.1.

[4] Dąbrowska G, Baum C, Trejgell A, and Hrynkiewicz K. Impact of arbuscular mycorrhizal fungi on the growth and expression of gene encoding stress protein–metallothionein BnMT2 in the non‐host crop Brassica napus L. J. Plant Nutr. Soil Sci. 2014;177(3):459-467. https://doi.org/10.1002/jpln.201300115.

Q3: Why you chose the four species to manipulate the experiment? Please give the reason.

RESPONSE: Thanks a lot for your comments. In our primary field investigations, the Gramineae species Setaria viridis vs. Arthraxon hispidus and Compositae Bidens pilosa vs. Bidens tripartita are successive pioneer species of karst communities as the herbaceous stage, which generally coexist in the same habitat as the main Gramineae and Compositae. In addition, A. hispidus and S. viridis are of the same family but different genera, while B. pilosa and B. tripartita have a common family and genera. Therefore, we also wanted to investigate whether AM fungi have different effects on different or the same taxonomic level of species. Of course, our results show that the biomass and nutrients of N and P were significantly different between A. hispidus and S. viridis of Gramineae, but not for B. pilosa and B. tripartita of Compositae under AMF.

Q4: In the discussion you paid more attention on the effect of AMF on plant growth and nutrient absorption. However, I think the combined effects of mycorrhizal fungi and indigenous microorganisms is more important to explanation.

RESPONSE: Thank you for your good suggestions. Yes, the combined effects of mycorrhizal fungi and indigenous microorganisms are more important to explain. AM fungi regulated plant growth positively affected by cooperating with indigenous microorganisms or negatively affected by competing with indigenous microorganisms. In the original manuscript in Line 354-387, we reviewed some literature about the combined effects of mycorrhizal fungi and indigenous microorganisms, including positive and negative to lead to our results, and discussed that the offset of the role of AM fungi by indigenous microorganisms may be caused by competition. In order to better explain, we added some literature to complement. See Line 364-365 and Line 367-371 of revised version.

Q5: Line 36: indigenous microbes are inconsistent with line 28.

RESPONSE: Thank you for your comments. We have corrected it in Line 36 of the revised version.

Q6: Line 106-108: do you sure the consistency of soil condition in physicochemical properties in AMF, AMI and CK? It is different in natural soil and sterilized soil in general cognition except for microbes.

RESPONSE: Thank you for your comments and questions. Here, we measured the soil quality, see the description of that the PH 8.2, total nitrogen (TN) 0.622 g, alkaline hydrolysis nitrogen (AN) 0.315 g, total phosphorus (TP) 1.274 g, available phosphorus (AP) 0.163 g, total potassium (TK) 37.79 g, and available potassium (AK) 0.532 g. Yes, autoclaving sterilization satisfied the requirements of sterilization, but affect some of the basic properties, including organic matter, specific surface area, PH, cation-exchange capacity, free iron/aluminum oxides and zero point of charge of the soils [1]. However, these basic properties affected by sterilization are not affected the research content of our article. In our experiment, we studied the role of indigenous microorganisms in affecting the growth and nutritional functions of plants regulated by AM fungi. In addition, there were strictly controlled experiments by the AMF and AMI (with AM fungus) and CK treatment (without AM fungus). Therefore, we pay more attention to chemical properties, and it is consistent in AMF, AMI and CK.

[1] Zhang H, Zhang J, Zhao B, Zhang C, and Zhang Y. Influence of autoclaving sterilization on properties of typical soils in China. Acta Pedologica Sinica. 2011;48(3):540-548. 

Q7: Line 118: 10 g Glomus mosseae should being 10 g Glomus mosseae inoculum.

RESPONSE: Thanks a lot for your comments. We have corrected it in Line 115 and Line 116 of the revised version.

Q8: Line 118-119: Did the 10 g Glomus mosseae inoculum include the spore, hyphal and root piece? Please give the information.

RESPONSE: Thank you for your comments and questions. Yes, the 10g Glomus mosseae inoculum includes the spore (above 100 spores per gram of soil), hyphae and colonized root pieces. For clarity, we deleted Line 145-147 in the original manuscript and added the information of 10 g Glomus mosseae inoculum in Line 121-123 of the revised version.

Q9: Line 152: are you sure this condition is the constant weight of drying?

RESPONSE: Thanks a lot for your comments. It is our negligence. We have checked carefully and corrected it in Line 148 of the revised version.

Q10: Line 173-174: Whether the data has been tested for normality and homogeneity of variance before analysis? In the best way, additional description is necessary to ensure the feasibility of statistical data.

RESPONSE: Thank you very much for your comments. Here in Statistical Analysis, we added the description by the sentence of “All of the data were tested for normality and homogeneity of variance before analysis”, see Line 170-171 of the revised version.

Q11: Line 154-156: how to calculate the accumulations, can you give us the details about it?

RESPONSE: Thank you for your comments and questions. In fact, we have given the details about the calculation of the accumulations in the original manuscript. Specifically, the nutrient concentrations of nitrogen and phosphorus of plant tissues of root and stem and leaf were determined. Further, the plant tissue accumulations of nitrogen and phosphorus were calculated respectively using nutrient concentration multiplying by biomass, then plant individual accumulations were accumulated by root and stem and leaf. Here, we revised the details about the calculation of accumulations. See Line 151-153 in the new version.

Q12: Line 290-291: Change "in negative N/P of " for "in negative N/P ratio of "; Change "in positive N/P of " for "in positive N/P ratio of ";

RESPONSE: Thank you for your comments. We already corrected it, see Line 288 and Line 289 of the revision.

Q13: Line 322: Change " the N/P of " for "the N/P ratio of "

RESPONSE: Thanks a lot for your comments. We have corrected it in Line 320 of the revised version.

Q14: Line 272-273: the indigenous microorganisms relieved the benefits of AM fungi on P accumulation. This sentence is unclear and contradicts the first part of the sentence (AM fungi improved P accumulation).

RESPONSE: Thank you for your comments. In fact, this is not contradictory. AM fungi can promote P accumulation in four karst pioneer species, however, indigenous microorganisms offset the benefits of inoculated AM fungi in promoting P accumulation. Of course, in order to express clearer, we corrected this sentence, see Line 271 of the revision.

Q15: Line 376: Change "streptomycetes " for " Streptomyces sp."

RESPONSE: Thank you for your comments. We already corrected it, see Line 378 of the revision.

Reviewer #3:

Q1: Soil microbial interactions play an important role for plant adaptation in natural habitat. As a kind of beneficial microorganisms, Arbuscular mycorrhizal fungi largely promote growth via the improvement of mineral nutrients for the host plant. This paper attempts to solve the interaction between AM fungi and indigenous microorganisms and explore the benefits of indigenous microorganisms on AM fungi promoting plant growth and nutrient utilization through four karst herbs, which were planted in three different microbial condition soil. The results indicated that the indigenous microorganisms relieved AM fungi's benefits in biomass and nutrient accumulation for plants. I believe this work is interesting and meaningful to apply mycorrhizal technology for restoring in degraded karst areas. However, it still needs to improve in some points as the potential publication of this paper, in detail as follows:

RESPONSE: Thank you very much for your comments. We have completely revised the manuscript in the new version. Further, according to the suggestions of Reviewer #1, we revised the title of the paper to ‘Indigenous microorganisms offset the benefits of growth and nutrition regulated by inoculated arbuscular mycorrhizal fungi for four pioneer herbs in karst soil’, and revised the conclusion in abstract, result and conclusion section.

Q2: Line 92-95: This sentence of “Thus, an experiment was ……with indigenous microorganisms”, is not necessary in the Introduction section. It is better to take it into the Methods section.

RESPONSE: Thanks a lot for your good suggestions. We have deleted Line 92-95 in the Introduction section in the original manuscript, and these contents have been presented in the Methods section in the original manuscript.

Q3: Line 103: do the “1120m.a.s.l” represent elevation? Please correct it.

RESPONSE: Thank you for your comments and questions. We have corrected it in Line 99 of the revised version.

Q4: Line 104: “soil microbial conditions” should be “soil microbial condition soil”.

RESPONSE: Thanks a lot for your comments. We already corrected it, see Line 100 of the revision.

Q5: Line 110: specify limestone soil as International Soil Classification

RESPONSE: Thank you for your good suggestions. The soil substrate was used by limestone in our experiment, according to your suggestion, we added the soil classification basis of FAO in Line 106, and changed the description of soil substrate.

Q6: Line 117-118: I confused the reason about promoting germination rate by yours treatment of 200g soil. Please check it and clear it.

RESPONSE: Thank you for your comments and questions. The three basic conditions for seed germination are appropriate temperature, appropriate water and sufficient air. Specifically, in our experiment, after covering the soil, the seeds can be kept germinating slowly and under a certain humidity. In order to express more clearly, we have modified this sentence, see Line 114 of the revision.

Q7: Line 121-122: This does not makes sense at all. Are you saying that you added AMF inoculum to your treatment control? If so, that does not constitute a control at all.

RESPONSE: Thank you for your comments and questions. In fact, we have given the details about added sterilized inoculum of Glomus mosseae to treatment control. The equal amount of sterilized inoculum and 10 ml of filtrate taken from sterilized inoculum were added in CK treatment, in order to maintain the consistency of microflora except for target fungi Glomus mosseae. Here, we revised the details about added sterilized inoculum of Glomus mosseae to treatment control. See Line 117-120 in the new version.

Q8: Line 121-123 This part (starting from "Especially, a 10 g…" and ending on "… a double-layer filter paper") is not clear at all. Please make it clear.

RESPONSE: Thanks a lot for your comments and suggestions. Regarding the long sentence, we have revised and made it clearer in the revised version, see Line 117-120.

Q9: Line 302: please correct the citation of He, Jiang et al.(2017).

RESPONSE: Thank you for your comments and questions. We already corrected it, see Line 300 of the revision.

Q10: Line 328: please correct the citation of Shen, Cornelissen et al.(2017). Check all reference citations in full text, I think it's not standard.

RESPONSE: Thank you for your comments and questions. We already checked and corrected all reference citations in full text, see Line 58, Line 60, Line 62, Line 71, Line 73, Line 78, Line 80, Line 162, Line 300, Line 309, Line 313, Line 326, Line 337, Line 359, Line 377, Line 380, Line 382 and Line 386 of the revision.

Q11: Line 332-335: This sentence was so long, I'm very confused with this result; please make it clear and shorten it.

RESPONSE: Thanks a lot for your comments and suggestions. Regarding the long sentence, we have revised and made it clearer and shorter in the revised version, see Line 330-332.

Q12: Lin 399: change “we can say that” being “we suggest that”, delete “Finally”.

RESPONSE: Thanks a lot for your good suggestions. We have deleted “Finally”, and have corrected it in Line 401 of the revised version.

Q13: The discussion needs further refinement and accuracy, comparing your results with previous researches for drawing relevant conclusions.

RESPONSE: Thanks a lot for your good suggestions. We have checked and revised the Discussion section carefully and deeply. On the whole, in order to make our points clear, the idea of revision was to explain and analyze the main research points directly, and then compare with others, as follows:

(1) AM fungi regulated plant growth positively affected by cooperating with indigenous microorganisms or negatively affected by competing with indigenous microorganisms. In the original manuscript in Line 354-387, we reviewed some literature about the combined effects of mycorrhizal fungi and indigenous microorganisms, including positive and negative to lead to our results, and drew relevant conclusions that the indigenous microorganisms offset the benefits of inoculated AM fungi in biomass and nutrient accumulation for pioneer plants in the karst habitat. 

(2) In order to better clarify, we added some literature to complement in Negatively section, we added the sentence ‘AM fungi can compete with indigenous microorganisms to produce different effects on plant growth, and clarified possible reason for this occurrence. ‘It is because indigenous microorganisms have great advantages in colonizing plant roots due to their priority in resources and allocating root space of the host plants compared with colonizers’, and the compare with ours. See the specific explanation of Line 364-365 and Line 367-371in the new version.

---

## [Decision Letter · Decision Letter 1]

28 Feb 2022

PONE-D-21-40761R1Indigenous microorganisms offset the benefits of growth and nutrition regulated by inoculated arbuscular mycorrhizal fungi for four pioneer herbs in karst soilPLOS ONE

Dear Dr. He,

Thank you for submitting your manuscript to PLOS ONE. After careful consideration, we feel that it has merit but does not fully meet PLOS ONE’s publication criteria as it currently stands. Therefore, we invite you to submit a revised version of the manuscript that addresses the points raised during the review process.

ACADEMIC EDITOR: The revised version has been improved a lot.  But the manuscript still has some problems as suggested by the reviewer.

We look forward to receiving your revised manuscript.

Kind regards,

Jian Liu

Academic Editor

PLOS ONE

Journal Requirements:

Reviewers' comments:

Reviewer's Responses to Questions

**Comments to the Author**

1. If the authors have adequately addressed your comments raised in a previous round of review and you feel that this manuscript is now acceptable for publication, you may indicate that here to bypass the “Comments to the Author” section, enter your conflict of interest statement in the “Confidential to Editor” section, and submit your "Accept" recommendation.

Reviewer #1: All comments have been addressed

Reviewer #2: All comments have been addressed

Reviewer #3: All comments have been addressed

2. Is the manuscript technically sound, and do the data support the conclusions?

Reviewer #1: Yes

Reviewer #2: Yes

Reviewer #3: Yes

3. Has the statistical analysis been performed appropriately and rigorously? 

Reviewer #1: Yes

Reviewer #2: Yes

Reviewer #3: Yes

4. Have the authors made all data underlying the findings in their manuscript fully available?

Reviewer #1: Yes

Reviewer #2: Yes

Reviewer #3: Yes

5. Is the manuscript presented in an intelligible fashion and written in standard English?

Reviewer #1: No

Reviewer #2: Yes

Reviewer #3: Yes

6. Review Comments to the Author

Reviewer #1: 1. Line 37-39. What is the purpose to compare the growth status of the species in this experiment? Is it essential for this topic?

2. Line 93-94. What were the evidences to support this hypothesis before this research was conducted?

3. Please add sub-headlines for the Discussion part. It is not clear what is the central topic for each paragraph. Still，there are too many discussions on the roles of AMF on plant growth, which were not the central topic of this study.

4. Line 396-399. The first argument is self-contradictory with the following statement.

5. Line 399-401. What is the significance of this finding?

6. Some grammar mistakes and English expressions are corrected in the tracked PDF.

Reviewer #2: Thanks for the authors. I think all the comments have been addressed so far. I have no other questions.

Reviewer #3: All comments were addressed. In this edition, the results and discussion were reorganized and now are clear for readers.

7. PLOS authors have the option to publish the peer review history of their article (what does this mean?). If published, this will include your full peer review and any attached files.

Reviewer #1: No

Reviewer #2: No

Reviewer #3: No

---

## [Author Response · Author response to Decision Letter 1]

3 Mar 2022

Journal Requirements

Q1: Please review your reference list to ensure that it is complete and correct. If you have cited papers that have been retracted, please include the rationale for doing so in the manuscript text, or remove these references and replace them with relevant current references. Any changes to the reference list should be mentioned in the rebuttal letter that accompanies your revised manuscript. If you need to cite a retracted article, indicate the article’s retracted status in the References list and also include a citation and full reference for the retraction notice.

RESPONSE: Thank you for your comments. According to the suggestions of Reviewer #1, we refined the discussion part and deleted five references [55], [56], [57] [69], [70] in the original manuscript, as follows:

[55] Tinker PB and Nye PH, Solute movement in the rhizosphere. 2000: Oxford University Press. 

[56] Leigh J, Hodge A, and Fitter AH. Arbuscular mycorrhizal fungi can transfer substantial amounts of nitrogen to their host plant from organic material. New Phytol. 2009;181(1):199-207. https://doi.org/10.1111/j.1469- 556

8137.2008.02630.x. 557

[57] Shao YD, Hu XC, Wu QS, Yang TY, Srivastava AK, Zhang DJ, et al. Mycorrhizas promote P acquisition of tea plants through changes in root morphology and P transporter gene expression. S. Afr. J. Bot. 2021;137:455-462. 

https://doi.org/10.1016/j.sajb.2020.11.028.

[69] Danuso F, Zanin G, and Sartorato I. A modelling approach for evaluating phenology and adaptation of two congeneric weeds (Bidens frondosa and Bidens tripartita). Ecol. Model. 2012;243:33-41. https://doi.org/10.1016/j.ecolmodel.2012.06.009 589

[70] Bartolome AP, Villaseñor IM, and Yang WC. Bidens pilosa L.(Asteraceae): botanical properties, traditional uses, phytochemistry, and pharmacology. Evid-Based Compl. Alt. 2013;2013. https://doi.org/10.1155/2013/340215.

Reviewer #1:

Q1: Line 37-39. What is the purpose to compare the growth status of the species in this experiment? Is it essential for this topic?

RESPONSE: Thank you for your comments. We checked and agreed with your view that it is not essential for this topic in the Abstract section, so we deleted Line 37-39 in the original manuscript.

Q2: Line 93-94. What were the evidences to support this hypothesis before this research was conducted?

RESPONSE: Thank you for your comments and questions.

In the revised version: (1) we revised the original summary between AM fungi and indigenous microorganisms to “Thus, the cooperation and competition between AM fungi and indigenous microorganisms are ineluctability in karst soil” See Line80-81 of the revision.

(2) we added the two sentences about previous studies as evidences to support the hypothesis, see Line 91-92 and Line 93-95 of the revision.

Q3: Please add sub-headlines for the Discussion part. It is not clear what is the central topic for each paragraph. Still，there are too many discussions on the roles of AMF on plant growth, which were not the central topic of this study.

RESPONSE: Thank you for your good suggestions. We added two sub-headlines for the Discussion part; see Line 297 and Line 340-341 of the revision. In addition, we further refined the Discussion section, please see the new version.

Q4: Line 396-399. The first argument is self-contradictory with the following statement.

RESPONSE: Many thanks for your comments and questions. We have checked and revised carefully, in order to express clearer, we modified “while the indigenous microorganisms offset the benefits of AM fungi foe host plants” to “However, the benefits from interaction were lower than benefits from AM, indicating that the indigenous microorganisms offset the benefits of AM fungi for host plants”. See Line 390 of the revision.

Q5: Line 399-401. What is the significance of this finding?

RESPONSE: Thanks a lot for your comments. We checked and agreed with your view that it is not essential for this topic in this manuscript, and we thought it is not the significance of this finding in this manuscript. Thus, we deleted Line 399-401 in the original manuscript.

Q6: Some grammar mistakes and English expressions are corrected in the tracked PDF.

RESPONSE: Thank you for your good suggestions. We further checked the language and refined expression, and modified some grammar mistakes and English expressions in the whole manuscript in the new revision according to your suggestions.

---

## [Editor Report · Decision Letter 2]

6 Mar 2022

PONE-D-21-40761R2Indigenous microorganisms offset the benefits of growth and nutrition regulated by inoculated arbuscular mycorrhizal fungi for four pioneer herbs in karst soilPLOS ONE

Dear Dr. He,

Thank you for submitting your manuscript to PLOS ONE. After careful consideration, we feel that it has merit but does not fully meet PLOS ONE’s publication criteria as it currently stands. Therefore, we invite you to submit a revised version of the manuscript that addresses the points raised during the review process.

ACADEMIC EDITOR: The revised version has been improved a lot. All the comments have been addressed. But the authors still need to polish the language and revise the language errors.

For example:

Line 390 : “the benefits form” should be “the benefits from”.

We look forward to receiving your revised manuscript.

Kind regards,

Jian Liu

Academic Editor

PLOS ONE

Journal Requirements:

Additional Editor Comments (if provided):

The revised version has been improved a lot. All the comments have been addressed. But the authors still need to polish the language and revise the language errors.

For example:

Line 390 : “the benefits form interaction” should be “the benefits from interaction”
---

## [Author Response · Author response to Decision Letter 2]

14 Mar 2022

Journal Requirements

Q1: The revised version has been improved a lot. All the comments have been addressed. But the authors still need to polish the language and revise the language errors.

For example:

Line 390 : “the benefits form interaction” should be “the benefits from interaction”

RESPONSE: Thank you for your comments and questions. We revised this sentence of “the benefits form interaction” being “the benefits from interaction”, see Line 383 of the revision. In addition, we further checked and modified the language and revised the language errors. Please see the new version.

---

## [Editor Report · Decision Letter 3]

23 Mar 2022

Indigenous microorganisms offset the benefits of growth and nutrition regulated by inoculated arbuscular mycorrhizal fungi for four pioneer herbs in karst soil

PONE-D-21-40761R3

Dear Dr. He,

We’re pleased to inform you that your manuscript has been judged scientifically suitable for publication and will be formally accepted for publication once it meets all outstanding technical requirements.

Kind regards,

Jian Liu

Academic Editor

PLOS ONE
---

## [Editor Report · Acceptance letter]

14 Apr 2022

PONE-D-21-40761R3 

Indigenous microorganisms offset the benefits of growth and nutrition regulated by inoculated arbuscular mycorrhizal fungi for four pioneer herbs in karst soil 

Dear Dr. He:

I'm pleased to inform you that your manuscript has been deemed suitable for publication in PLOS ONE. Congratulations! Your manuscript is now with our production department. 

Kind regards, 

on behalf of

Dr. Jian Liu 

Academic Editor

PLOS ONE